# Peroxisomal Cofactor Transport

**DOI:** 10.3390/biom10081174

**Published:** 2020-08-12

**Authors:** Anastasija Plett, Lennart Charton, Nicole Linka

**Affiliations:** Institute of Plant Biochemistry and Cluster of Excellence on Plant Sciences (CEPLAS), Heinrich Heine University, Universitätsstrasse 1, 40,225 Düsseldorf, Germany; Anastasija.Plett@hhu.de (A.P.); Lennart.Charton@hhu.de (L.C.)

**Keywords:** peroxisomes, metabolism, carrier, cofactor

## Abstract

Peroxisomes are eukaryotic organelles that are essential for growth and development. They are highly metabolically active and house many biochemical reactions, including lipid metabolism and synthesis of signaling molecules. Most of these metabolic pathways are shared with other compartments, such as Endoplasmic reticulum (ER), mitochondria, and plastids. Peroxisomes, in common with all other cellular organelles are dependent on a wide range of cofactors, such as adenosine 5′-triphosphate (ATP), Coenzyme A (CoA), and nicotinamide adenine dinucleotide (NAD). The availability of the peroxisomal cofactor pool controls peroxisome function. The levels of these cofactors available for peroxisomal metabolism is determined by the balance between synthesis, import, export, binding, and degradation. Since the final steps of cofactor synthesis are thought to be located in the cytosol, cofactors must be imported into peroxisomes. This review gives an overview about our current knowledge of the permeability of the peroxisomal membrane with the focus on ATP, CoA, and NAD. Several members of the mitochondrial carrier family are located in peroxisomes, catalyzing the transfer of these organic cofactors across the peroxisomal membrane. Most of the functions of these peroxisomal cofactor transporters are known from studies in yeast, humans, and plants. Parallels and differences between the transporters in the different organisms are discussed here.

## 1. Introduction

Peroxisomes are eukaryotic organelles that are surrounded by a single lipid bilayer membrane [1,2]. They fulfil a range of metabolic functions, which are essential for development and cellular signaling. Depending on the organism, cell type, growth, and environmental conditions, peroxisomes participate in the detoxification of reactive oxygen/nitrogen species, β-oxidation of fatty acids, synthesis of plasminogen, isoprenoids, penicillin, phylloquinone, glycine betaine, biotin and hormonal signal molecules, catabolism of purines, polyamines, amino acids, and methanol, as well as in the glyoxylate cycle, pentose phosphate pathway, and photorespiration [1,2]. Currently, the list of peroxisomal tasks appears to be far from exhaustive.

The multiple roles of peroxisomes depend on the functional interplay with other organelles. A large number of chemically diverse metabolites consumed and released by the peroxisomes have to be exchanged between subcellular compartments [3,4,5,6]. The transfer of metabolites across the peroxisomal membrane has been a controversial scientific debate for several decades. Initial studies on isolated peroxisomes suggested that the peroxisomes are freely permeable. However, this in vitro permeability was later withdrawn and explained by disruptions of protein/membrane structures during peroxisome isolation procedure. The current consensus is that transport proteins are responsible for the transfer of solutes into and out of peroxisomes [3,4,5,6]. Electrophysiological experiments using purified peroxisomal membranes revealed the existence of nonspecific porin-like channels that allow the free diffusion of low-molecular weight compounds (<300 Da) with a broad substrate specificity [7,8]. In addition, genetic mutant analyses discovered genes encoding for specific carrier proteins that catalyze the flux of larger hydrophilic solutes, like cofactor molecules, such as ATP, Coenzyme A (CoA), and NAD [9,10,11,12,13].

The transport of these cofactors has a crucial impact on peroxisome function. For example, ATP, CoA, and NAD are required for peroxisomal fatty-acid activation and oxidation via β-oxidation, which is a conserved metabolic pathway in yeast, mammals, humans, and plants [3,14,15]. These cofactor molecules are synthesized outside peroxisomes. Due to their size and charge, they cannot pass the lipid bilayer by free diffusion. Thus, peroxisomal transport proteins are mandatory to replenish the cofactor demand of the peroxisomal enzymes [3,14,15]. Notably, peroxisomes offer an alternative route for cofactor transport. In contrast to mitochondria and chloroplasts, peroxisomes lack a protein synthesis machinery and have the capacity to import folded and even oligomeric proteins [16,17,18]. It was assumed that the import of tightly bound cofactors, such as flavin adenine dinucleotide (FAD) and thiamine pyrophosphate (TPP), could be coupled to protein transport, as is the case for the Tat system of bacteria and chloroplasts, which export folded proteins to the periplasm or the thylakoid lumen, respectively [19]. For instance, the FAD-dependent alcohol oxidase and acyl-CoA oxidase bind their cofactor FAD in the cytosol and are then imported as fully folded holo-enzymes into yeast peroxisomes [20,21]. Such a cofactor-coupled protein import mechanism was also reported for the mammalian 2-hydroxyacyl-CoA lyase, a TPP-dependent enzyme [22]. However, specific cofactor carrier proteins for ATP, CoA, and NAD are essential to generate and maintain a physiologically relevant peroxisomal pool of free cofactor molecules, which are essential for an efficient and functional metabolism.

Members of the mitochondrial carrier family (MCF) that mediate the transport of a wide range of organic cofactors into peroxisomes have been discovered in diverse eukaryotes [23,24,25,26,27]. This eukaryotic group of membrane transport proteins corresponds to the 2.A.29 family according to the Transporter Classification Database (TCDB) and is named in mammals solute carrier family 25 (SLC25) [28]. MCF is a large family of proteins with about 30 members in yeast and more than 50 in humans and plants. Although the name of this family suggests that they are exclusively located to mitochondria, several members are present in other organelles, such as peroxisomes, endoplasmic reticulum, chloroplasts, and plasma membrane [23,24,26,27,29]. Despite their conserved basic structure composed of three repetitive modules, MCF proteins are highly diverse in terms of substrate specificity and transport mode. They mediate the transport of a large variety of solutes that differ in size and nature, such as protons, inorganic ions, inorganic form of “phosphate”, carboxylic acids, amino acids, and nucleotides. In most cases, MCF members mediate a strict counter-exchange but also operate as a uniporter or symporter [26,27,29,30]. These features suggest that this protein family was most likely exploited as a valuable basis for a fast establishment of a subset of carriers with a broad range of different transport functions in the cell during eukaryotic evolution.

This review deals with MCF proteins that are known to be peroxisomal cofactor carriers in budding yeast *Saccharomyces cerevisiae*, humans, and the model plant *Arabidopsis thaliana* and highlights the recent progress on their biochemical and physiological function for the peroxisomal metabolism.

## 2. Cofactor Transport for Yeast Peroxisomes

The main metabolic function of peroxisomes in *S. cerevisiae* is the degradation of fatty acids via β-oxidation to use these compounds as carbon and energy source [31]. The pathway depends on the availability of ATP, CoA, and NAD in the peroxisomal matrix (Figure 1). The uptake of fatty acids into peroxisomes occurs via two routes depending on the fatty-acid chain length [32]. Small- and medium-chain fatty acids (C4–12) are transported as free fatty acids through passive diffusion, while long-chain fatty acids (C14–20) are delivered as acyl-CoA esters by a peroxisomal ATP-binding cassette (ABC) transporter Pxa1p-Pxa2p [33,34]. However, during this transport process the CoA moiety is cleaved off. Thus, both entry pathways for fatty acids lead to the delivery of nonesterified fatty acids [35]. However, prior to peroxisomal β-oxidation, these free fatty acids must be activated to acyl-CoA esters. This intraperoxisomal esterification is catalyzed by a peroxisomal acyl-CoA synthetase Faa2p and/or a bifunctional fatty-acid transporter Fat1p at the peroxisomal membrane and requires ATP and CoA [35]. Once the acyl-CoA esters are fed into the peroxisomal β-oxidation cycle, their oxidative degradation requires NAD as an electron acceptor [31].

In 2001, two groups independently identified one MCF member from *S. cerevisiae* as a peroxisomal ATP carrier, which was named Ant1p for Adenine nucleotide transporter 1 [9,10]. Disruption of Ant1p results in yeast cells that exhibit an impaired growth in the presence of medium-chain fatty acids, such as lauric acid, as the sole carbon source. In vivo activity of the ATP-consuming firefly luciferase, targeted to *ant1*Δ peroxisomes, was strongly reduced, implying a depleted peroxisomal ATP content in the intact mutant cells [36]. Gene expression analysis revealed the presence of oleate response elements in the promoters of ANT1 and other β-oxidation genes, which are responsible for an induced gene expression via the Pip2p-Oaf1p transcription factor when grown in the presence of the long-chain fatty acid, oleic acid [36]. These observations led to the hypothesis that Ant1p is necessary for metabolizing fatty acids via peroxisomal β-oxidation as a carbon and energy source [9,10].

Direct transport studies with purified Ant1p protein provided conclusive evidence for the role of Ant1p as an ATP transporter [9]. *Escherichia coli* has been a suitable system for expression, purification, and subsequent functional reconstitution into liposomes. In vitro uptake studies using diverse nucleotides as substrates demonstrated that Ant1p specifically catalyzes the transport of ATP, ADP, and AMP [9]. However, the protein does not accept other ATP-related molecules, such as CoA or NAD, as transport substrates [37]. Another unique characteristic of Ant1p is that it exhibits two transport modes. It mediates not only the exchange but also catalyzes the uniport of adenine nucleotides [9].

Based on its transport features, two physiological functions for Ant1p can be concluded: (1) It facilitates the uptake of cytosolic ATP in a unidirectional mode for loading peroxisomes with ATP early in their genesis. (2) During high rates of β-oxidation, ATP is directly consumed by the ATP-dependent fatty-acid activation, releasing high amounts of AMP. Ant1p ensures the counter-exchange of ATP into peroxisomes against AMP to avoid accumulation of the latter molecule in the peroxisome, which would, on the other hand, deplete the nucleotide pool in the cytosol [9,10]. However, the loss of the Ant1p in *S. cerevisiae* did not fully abolish fatty-acid oxidation activity [10]. The *ant1*Δ mutant was still able to degrade lauric acid and oleic acid. Moreover, 20% of the peroxisomal luciferase activity was still detectable, indicating the presence of low ATP levels in *ant1*Δ peroxisomes [10]. In contrast, a complete block of β-oxidation rates for medium- and long-chain fatty acids was observed in yeast cells lacking the two ATP-dependent acyl-CoA synthetases Faa2p and Fat1p [35]. Since no other ATP-generating systems have been discovered so far, future research will address how an alternative bypass route provides peroxisomes with ATP particularly for fatty-acid metabolism.

Very little is known about the uptake of CoA and NAD by yeast peroxisomes. In yeast, cytosolic CoA probably enters the peroxisomal matrix via the Pxa1p–Pxa2p transporter, which cleaves off the CoA moiety from the imported acyl-CoA ester. Alternatively, a specific transport protein might facilitate the CoA uptake. Inside peroxisomes CoA is not only essential for the fatty-acid activation but also for the thiolytic cleavage within the β-oxidation cycle via the action of acyl-CoA thiolases [31]. The CoA bound to the acyl chain is released when acetyl-CoA, the product of β-oxidation, is exported via the carnitine shuttle or enters the glyoxylate cycle. In order to regulate the CoA homeostasis, the peroxisomal CoA diphosphatase Pcd1p hydrolyzes CoA to adenosine 3′,5′-diphosphate and 4′-phospho-pantetheine [38]. The resulting products need to exit the peroxisomes to enter the cytosolic CoA salvage pathway. Both CoA derivatives might function as potential counter-exchange substrates for the peroxisomal CoA importer. However, it is currently unclear whether yeast peroxisomes harbor such a CoA transport protein.

An additional cofactor uptake system must exist in yeast for loading the peroxisomal lumen with NAD. During the β-oxidation cycle, NAD is reduced to NADH, which is directly re-oxidized by the peroxisomal malate dehydrogenase Mdh3p, an important component of the malate–oxaloacetate shuttle [39]. By action of this shuttle, peroxisomal NADH is indirectly transported to mitochondria, where it is re-oxidized to NAD, and then returned back to peroxisomes. Genetic in vivo studies with *mdh3*Δ mutants, which were unable to metabolize fatty acids, demonstrated that the peroxisomal membrane of *S. cerevisiae* is impermeable to NAD(H) [39]. This indicated that a direct exchange of NAD against NADH across the peroxisomal membrane does not occur in yeast, and thus the transfer of reducing equivalents is mediated by NAD-linked redox shuttles [40]. In addition, yeast possess additional redox shuttles to maintain the intraperoxisomal redox balance, such as the glycerol-3-phosphate/dihydroxyacetone phosphate NAD-linked shuttle and the 2-oxoglutarate/isocitrate NADP-linked shuttle [40,41]. A prerequisite for redox metabolism is a constant concentration of NAD(H) inside peroxisomal lumen. The NAD(H) homeostasis is achieved by the peroxisomal NADH diphosphatase Npy1p [42], converting NAD(H) to AMP and nicotinamide mononucleotide. In order to recycle these products of NAD hydrolysis in the cytosol, they have to be exported by a specific carrier, which might be coupled to the import of NAD. However, the mechanism to initially generate an NAD pool inside peroxisomes is still unknown.

Members of the MCF have been identified to mediate an efficient subcellular distribution of ATP, CoA, and NAD within the eukaryotic cell. *S. cerevisiae* contains 35 MCF-type proteins, including the mitochondrial CoA carrier Leu5p [43] and the mitochondrial NAD carriers Ndt1p and Ndt2p [44]. Two scenarios are possible: (1) One of the so far uncharacterized MCF or even non-MCF proteins in yeast might catalyze the peroxisomal cofactor uptake. (2) The mitochondrial CoA and/or NAD transporter might be dually localized to mitochondria and peroxisomes to adapt to the cellular needs. Future analyses will discover whether and which carrier-type protein might mediate the peroxisomal cofactor transport.

## 3. Cofactor Transport for Human Peroxisomes

In humans, peroxisomes are present in all cell types, except in erythrocytes [1,45,46]. The pivotal role of these organelles is emphasized by a variety of severe genetic diseases linked to peroxisome dysfunction. Most of these disorders are caused by mutations in genes coding for peroxisomal enzymes involved in metabolic pathways [45]. In humans, the key metabolic function of human peroxisomes is β-oxidation. While mitochondrial β-oxidation handles the bulk of dietary fatty acids, such as palmitic acid and oleic acid, the peroxisomal β-oxidation plays a crucial role in the degradation of a more diverse spectrum of carboxylic acids, including long-chain fatty acids (LCFAs), very long-chain fatty acids (VLCFAs, >C22), branched-chain fatty acids (e.g., pristanic and phytanic acids), and long-chain dicarboxylic acids. In addition to its catabolic functions, peroxisomal β-oxidation is involved in the biosynthesis of the bile acid intermediates di- and tri-hydroxycholestanoic acid and the essential omega-3-fatty acid docosahexaenoic acid (C22:6 n-3), a primary structural component of the human brain, cerebral cortex, skin, and retina [1,45,46].

To import the diverse carboxylic acids into peroxisomes, three half-size ABC transporters of the subfamily D (ABCD) reside in the peroxisomal membrane [47,48,49]. They function mainly as homodimers with partially overlapping substrate specificities. ABCD1 (ALDP) has a higher affinity for saturated VLCFAs, whereas ABCD2 (ALDR) transports shorter and (poly)unsaturated VLCFAs [50,51]. In contrast, ABCD3 (PMP70) imports branched-chain fatty acids, long-chain dicarboxylic acids, and bile acid intermediates into human peroxisomes [52,53]. The peroxisomal ABCD proteins transport their substrates as CoA esters, whereas the peroxisomal membrane-bound acyl-CoA binding protein ACBD5 functions as a cytosolic receptor for VLCFA-CoAs and passes them on to the VLCFA transporter ABCD1 [54]. Furthermore, ABCD1–3 share the same transport mode with the yeast fatty-acid transporter described in the previous chapter [55,56]. An intrinsic acyl-CoA thioesterase activity couples the translocation step to the hydrolysis of the CoA ester, leaving a free acid in the peroxisomes that must be re-activated with CoA for β-oxidation [57]. The human genome encodes several different acyl-CoA synthetases, catalyzing the ATP-dependent activation of fatty acids and related compounds to acyl-CoA in the presence of ATP [58,59]. Few isoforms have been reported to be linked to the peroxisomal membrane, but whether these membrane-associated proteins are active inside or outside the peroxisome has been the subject of recent debates [58,59]. Still, the active site of one human acyl-CoA synthetase has been located to the peroxisomal matrix. It is assumed that the enzyme specifically activates peroxisomal pristanic acid produced by α-oxidation of phytanic acid [60]. Reports on the human ABCD1 suggest that the import of VLCFA-CoA esters into peroxisomes is not dependent on peroxisome internal activation [61]. Still, the complementation of the yeast *pxa1/pxa2Δ* double mutant with the human ABCD1 requires the presence of the peroxisomal acyl-CoA synthetase Faa2p for the activation of the β-oxidation substrates [35]. Future research will resolve these partly contradictory hypotheses.

In the case that β-oxidation substrates enter peroxisomes as free fatty acids, a pool of ATP and CoA is needed for the peroxisomal re-esterification (Figure 2). Consequently, human peroxisomes have to be supplied with both cofactor molecules, if CoA is released in the cytosol by ABCD proteins [55,56]. The resulting acyl-CoA esters are then fed into β-oxidation, which depends on peroxisomal NAD as an electron acceptor. The last step of this pathway, catalyzed by the acyl-CoA thiolase, uses free CoA to cleave off one acetyl-CoA molecule from the acyl-chain [46]. The shortened acyl-CoA ester undergoes additional cycles but will not be completely degraded. Thus, the peroxisomal β-oxidation generates several different medium-chain acyl-CoAs, besides propionyl-CoA and acetyl-CoA. These products are then exported via the peroxisomal carnitine shuttle from peroxisomes to mitochondria for further metabolism [46]. This export mechanism releases CoA in the peroxisomal lumen, which can be recycled for the thiolytic cleavage during β-oxidation. Peroxisomal thioesterase may also function in exporting the products of β-oxidation by releasing them from CoA esters [62].

In humans, one member the MCF has been discovered as peroxisomal membrane protein of 34 kDa (PMP34) [63]. This human MCF protein is classified as member 17 of the solute carrier family 25 (SLC25A17). Due to its high homology to the yeast peroxisomal ATP carrier, it was hypothesized that human SLC25A17 is a functional ortholog of Ant1p [63,64]. To address this, human SLC25A17 was expressed in the *ant1*Δ yeast mutant [64]. While peroxisomal β-oxidation activity in *ant1*Δ cells were only 20% of wild type cells, the fatty-acid degradation was restored to approximately 60% of wild type in *ant1*Δ, expressing the human SLC25A17. This partial rescue of the *ant1*Δ phenotype with human SLC25A17 suggested a role in providing peroxisomes with ATP for fatty-acid oxidation [64]. Functional reconstitution of yeast expressed and purified SLC25A17 protein in lipid vesicles revealed detectable ATP import activity across the liposomal membrane [64]. Both observations led to the conclusion that the human SLC25A17 functions as an adenine nucleotide transporter, catalyzing an exchange of adenine nucleotides across the peroxisomal membrane [64]. The group of Ferdinando Palmieri repeated the uptake assays with the human SLC25A17 recombinantly expressed in *E.coli* and purified by affinity chromatography [65]. Surprisingly, it exhibited extremely low uptake rates of ATP, which might explain the partial complementation of the *ant1*Δ mutant with the human SLC25A17 [64]. In contrast, high transport activities of recombinant SLC25A17 were discovered for AMP exchange against CoA, dephospho-CoA, and FAD. It also catalyzes the AMP import against internal FMN, ADP, adenosine 3′,5′-diphosphate (PAP), and to a lesser extent, NAD [65]. Considering lower K_M_ and K_i_ values suggested a higher affinity of SLC25A17 for CoA, AMP, FAD, and FMN as substrates than for NAD, ADP, and ATP [65]. The physiological role of such a cofactor carrier with a versatile transport functions remains unclear for human peroxisomes.

The in vivo function of SLC25A17 has been further investigated in genetic mutants of orthologs in the model organisms zebrafish (*Danio rerio*) and mice (*Mus musculus*) [66,67]. The zebrafish genome contains two Slc25a17 proteins, which have been both simultaneously down-regulated by a morpholino-based antisense approach [66]. During the first four days of development, zebrafish embryos rely entirely on its nutrient-rich yolk sac to sustain growth and survival. In this phase, peroxisomal β-oxidation supports the utilization of very long-chain fatty acids, as well as the synthesis of plasmalogens, providing energy and structural cellular components [68]. Silencing of the two *slc25a17* genes at 3–4 days post-fertilization led to an accumulation of very long-chain fatty acids and a reduction of ether-phospholipids in the zebrafish embryos [66]. This altered cellular lipid composition caused a severe failure in the development of multiple organs, including the swim bladder. To test if the loss of peroxisomal CoA or NAD impairs β-oxidation function, these cofactors were co-injected together with the *slc25a17*-specific antisense oligomers into zebrafish embryos. Exogenously supplied CoA efficiently rescued the defective swim bladder, whereas NAD co-injection failed to restore the developmental defects associated with *slc25a17* knockdown [66]. In vitro uptake experiments demonstrated that both Slc25a17 proteins function redundantly with a preference towards CoA, instead of NAD and ATP, similar to the human carrier [66]. These findings suggest that Slc25a17 and Slc25a17-like functions additively as CoA transporters, which are involved in peroxisomal lipid metabolism and are thus essential for normal embryonic growth in zebrafish [66].

Mice lacking the SLC25A17 carrier by insertional mutagenesis did not show any obvious phenotype [67]. In particular, the diverse functions of the peroxisomal β-oxidation were not compromised in the SLC25A17-deficient mice. Only the degradation of phytol-derived branched-chain fatty acids was considerably impaired. Phytol, a constituent of chlorophyll, is converted to phytanic acid [69,70,71]. This methyl-branched fatty acid first activated by the long-chain acyl-CoA synthetases ACSL1/4 in the cytosol and then imported as acyl-CoA ester via ABCD3 transporter into peroxisomes [58,59], where it enters the peroxisomal α-oxidation. This pathway produces pristanic acid inside peroxisomes. This fatty acid is activated to pristanoyl-CoA catalyzed by the peroxisomal very-long-chain acyl-CoA synthetase ACSVL1 [60] for further conversion by multiple rounds of peroxisomal β-oxidation. The resulting medium-chain fatty acids are then completely oxidized by the mitochondrial β-oxidation [69,70,71]. Notably, pristanic acid is also present in human diets, and for its degradation via the peroxisomal β-oxidation, it is activated in the cytosol and imported as acyl-CoA ester via the ABCD3 transporter into peroxisomes [69,70,71].

Upon phytol feeding, the SLC25A17 knockout mice accumulated phytanic and pristanic acid as well as their CoA-esters in the liver, resulting in an enlarged organ and hepatic inflammation [67]. These abnormalities of the phytol-fed knockout mice suggested that the phytol degradation process depends on the peroxisomal cofactor supplied by SLC25A17. Unfortunately, the transport function of mice SLC25A17 has not yet been characterized, and thus we can only hypothesize about its preference for ATP, CoA, or NAD as a substrate [67]. Assuming that mouse SLC25A17 functions as a peroxisomal CoA carrier, as postulated for the zebrafish ortholog, the specific phenotype of the slc25a17-deficient mice might be caused by a shortage of peroxisomal CoA. Since the peroxisomal activation of pristanic acid as well as the thiolytic cleavage during β-oxidation of the resulting medium-chain fatty acids demand free CoA in the peroxisomal matrix, the levels of both phytol-derived fatty acids and acyl-CoA esters were elevated in the liver of the KO mice after phytol feeding [67].

In general, the activity of the acyl-CoA thioesterases prevent CoA deficiencies in peroxisomes from humans and mice [62]. These peroxisomal enzymes hydrolyze CoA esters, resulting in CoA formation, and thus can regulate the peroxisomal CoA pool [62]. In order to ensure a net CoA influx, the mice SLC25A17 carrier might catalyze the import of two CoA molecules against one molecule of adenosine 3′,5′-diphosphate (PAP) and 4′-phosphopantetheine (4′-PP). Both counter-exchange substrates are products of the hydrolysis of one CoA molecule, provided by peroxisomal Nudix hydrolases [72,73,74]. For the human SLC25A17, it was shown that it transports PAP, however 4′-PP as a potential substrate has not been tested in this work [65].

Another open question is why the CoA-dependent activation of other β-oxidation substrates is not impaired in the SLC25A17-deficient mice. The mild phenotype of the loss-of-function mutant, which can only be unmasked upon phytol treatments, might point to an alternative way to provide peroxisomes with CoA. For instance, another CoA carrier protein may exist in the peroxisomal membrane or an alternative splicing variant could lead to dual targeting of the human mitochondrial CoA carrier [75,76]. In addition, it cannot be excluded that peroxisomal CoA-dependent enzymes are imported together with their cofactor, as it was described for FAD-dependent enzymes previously [20,21]. Nevertheless, this study may be applicable to the human system, indicating that SLC25A17 deficiency in humans is unlikely to be lethal but could cause an impaired metabolization of branched-chain fatty acids in older adults [67].

Up to now, the SLC25A17 proteins are the only MCF-type cofactor carrier known to be present in peroxisomes of humans, mice, and zebrafish. Orthologs of the peroxisomal ATP carrier Ant1p from yeast or the peroxisomal NAD carrier from Arabidopsis are required to provide metabolic pathways with ATP and NAD. Beside β-oxidation, ATP and NAD are also involved in different peroxisome-associated processes in humans and mice. For instance, intraperoxisomal ATP can be used for the post-translational phosphorylation of peroxisomal enzymes by protein kinases [77], proper folding of peroxisomal matrix proteins [78], removal of oxidatively damaged, and misfolded proteins via ATP-stimulated proteases [79,80]. NAD, in particular the phosphorylated, reduced form NADPH, is also needed for the oxidation of unsaturated fatty acids [46,70], as well as for the antioxidants defense systems, such as the ascorbate–glutathione cycle, to detoxify reactive oxygen species, such as the glutathione cycle [81]. Diverse peroxisomal NAD(P)-redox shuttle mechanisms, such as malate/oxaloacetate, lactate/pyruvate, and 2-oxoglutarate/isocitrate-based shuttle systems, regenerate NAD or NADPH and thus participate in regulating the redox homeostasis in human peroxisomes [4]. However, it remains to be investigated whether specific transport proteins are involved in the provision of ATP and NAD for peroxisomes in humans and animals.

## 4. Cofactor Transport for Plant Peroxisomes

In contrast to humans and animals, peroxisomes are the sole site of β-oxidation in plants. This essential metabolic pathway is involved in diverse developmental and signaling processes, participating in fatty-acid catabolism and the biosynthesis of several major phytohormones, including jasmonic acid, indole-3-acetic acid (auxin), and salicylic acid [2,82,83].

Another important function of β-oxidation is the mobilization of storage oil in oil-seed species, such as the model plant *Arabidopsis thaliana* [84,85]. These plants store energy in form of triacylglycerol in the seeds in order to secure survival of the next generation. During germination the embryos utilize the storage oil to enable the seedling growth and development, until it becomes photoautotrophic. In Arabidopsis, the long-chain fatty acids that are hydrolyzed from reserve lipids are imported into peroxisomes for further degradation by β-oxidation [84,85]. The import of fatty acids as CoA esters is mediated by the peroxisomal ABC transporter, which was named COMATOSE (CTS) after its dormancy phenotype in Arabidopsis [86,87,88,89]. The CTS transport mechanism is the same as described for the peroxisomal ABC transport proteins from yeast and human. The CoA-moiety is cleaved off during the import, which necessitates the re-esterification of fatty acids to CoA prior to entering β-oxidation [90]. This activation reaction is catalyzed by two peroxisomal long-chain acyl-CoA synthetases LACS6 and LCAS7 in an ATP-consuming reaction [91]. In the absence of both enzymes in peroxisomes, Arabidopsis seedlings were compromised in storage-oil mobilization, leading to an arrested seedling growth shortly after germination [91]. This phenotype implies that the activation of the fatty acid inside plant peroxisomes is essential for their breakdown. The resulting acyl-CoAs are then degraded by the NAD-dependent β-oxidation cycle [84,85]. The last step of this pathway uses free CoA for the thiolytic cleavage of the acyl-CoA, generating acetyl-CoA and a shortened acyl-CoA. The latter can re-enter the β-oxidation pathway for further breakdown. The resulting acetyl-CoA is fueled into the peroxisomal glyoxylate cycle, producing four-carbon metabolites that can be converted to sucrose as a carbon and energy source [84,85]. With respect to the cofactor input, the peroxisomal enzymes of fatty-acid degradation have the same demand of ATP, CoA, and NAD in humans, animals and yeast [3,92] (Figure 3).

On the basis of amino-acid-sequence similarity to the peroxisomal ATP carrier Ant1p from *S. cerevisiae*, two Arabidopsis MCF members have been identified to be localized to the peroxisomal membrane [12,13]. These plant carriers were independently able to suppress the β-oxidation phenotype of the ant1Δ yeast mutant [13], indicating that both proteins import ATP into yeast peroxisomes. A biochemical characterization using recombinant PNC proteins revealed that the ATP import was mediated in a strict counter exchange transport mode with ADP or AMP, similar to the yeast ortholog [12,13]. Whereas substrate specificity was similar to the yeast ortholog, none of the PNC proteins showed uniport transport activity. Due to their restricted substrate spectrum, the Arabidopsis MCF members were annotated as peroxisomal adenine nucleotide carriers PNC1 and PNC2. Arabidopsis mutants were generated in which PNC1 and PNC2 were simultaneously silenced using RNA interference to analyze their in vivo role [12,13]. Since the import of cytosolic ATP into peroxisomes is essential for β-oxidation, the seedlings of the RNAi lines were compromised in seedling establishment [12,13]. The arrested seedling phenotype was caused by a block in storage-oil mobilization, resulting in an accumulation of storage-oil-derived fatty acids and acyl-CoA esters. This result suggests that peroxisomal ATP uptake mediated by PNC1/2 is critical for utilizing fatty acids through β-oxidation to fuel seedling growth [12,13]. It is hypothesized that PNC-mediated ATP import occurs in exchange with peroxisomal AMP, which is released during the fatty-acid activation step by LACS6/7 [91]. In addition, other metabolic pathways that are linked to peroxisomal β-oxidation were also affected in these *pnc1/2* RNAi lines, such as biosynthesis of phytohormones, the catabolism of membrane lipids during dark-induced senescence, and the degradation of branched-chain amino acids (unpublished work). Together, these findings emphasize that the peroxisomal ATP carriers are the primary source of peroxisomal ATP, meaning no other major ATP-generating systems, such as substrate-level phosphorylation, exist in Arabidopsis peroxisomes.

Our knowledge about other ATP-consuming reactions in plant peroxisomes beside β-oxidation is limited. Recently, it was demonstrated that the ATP-dependent enzymes that catalyze the last steps of the cytosolic mevalonate pathway for the synthesis of isopentenyl diphosphate are located inside plant peroxisomes [93,94]. Peroxisomal ATP might also be crucial for the regulation of protein function. Phospho-proteomic studies revealed that many enzymes involved in photorespiration are regulated via post-translational phosphorylation [95,96,97]. Accordingly, several protein kinases have also been identified in plant peroxisomes, but their target proteins still need to be discovered [98,99]. A role of the PNC proteins as ATP/ADP carrier remains to be tested.

In order to support NAD-dependent fatty-acid oxidation during early seedling growth, the import of NAD into Arabidopsis peroxisomes is mediated by the peroxisomal NAD carrier, called PXN [100,101]. PXN belongs to the MCF and represents, regardless of the species, the first, and so far, only discovered peroxisomal carrier for NAD. In vitro uptake experiments of reconstituted recombinant protein discovered that PXN can transport many organic cofactors and related molecules, including NAD, NADH, ADP, AMP, and CoA, in an antiport mode [100,101]. The outcome of the biochemical (in vitro) assays suggest versatile transport functions for PXN, catalyzing the import of NAD or CoA against AMP or the exchange of NAD/NADH. However, complementation studies using different yeast mutants restricted the in vivo function of PXN to provide peroxisomes with cytosolic NAD in exchange with intraperoxisomal AMP [37]. Since the peroxisomal malate/oxaloacetate shuttle is involved in the export of NADH from peroxisomes and the import of re-oxidized NAD [102,103], PXN as an NAD/AMP exchanger might function to build up and/or replenish the peroxisomal NAD pool [14,37,101]. For instance, a net NAD import is facilitated, if the exported AMP is strictly accompanied by a unidirectional AMP import to balance the loss of peroxisomal AMP. Such an adenylate uniporter has only been described for chloroplasts so far [104,105]. In another scenario, PXN might transport two NAD molecules. One NAD is used for the peroxisomal metabolism, the other is hydrolyzed to AMP and reduced nicotinamide mononucleotide by the peroxisomal Nudix hydrolase in Arabidopsis [106,107]. Both hydrolysis products might counter-exchange during the uptake of two NAD molecules [14,101].

The role of PXN in providing the peroxisomal β-oxidation with NAD implies that a deletion of this gene would affect storage-oil mobilization, which is essential for seedling establishment. Surprisingly, Arabidopsis *pxn* knockout mutants did not display any obvious seedling phenotype, as described for the *PNC1/2* RNAi lines [100,101]. The fatty-acid composition in these mutant seedlings revealed that fatty-acid breakdown was delayed during storage-oil turnover, indicating that peroxisomal NAD import mediated by PXN contributes to an optimal operation of storage-oil degradation [100,101]. Recently, PXN was identified to be involved in photorespiration under fluctuating and high-light conditions [108]. The decreased activity of the photosystems in the *pxn* plants could be rescued by elevated CO_2_ concentrations, which represses the flux through the photorespiratory pathway. The authors proposed that PXN can supply plant peroxisomes with the increased demand of NADH during photorespiration [108]. Both defined phenotypes point to an alternative mechanism to fuel plant peroxisomes with NAD; most likely by a redundant uptake system, that is either mediated by a specific carrier, protein-coupled NAD import, or by fusion of NAD-preloaded pre-peroxisomes derived from the ER [14].

Plant peroxisomes have to control their peroxisomal CoA pool, since this cofactor is crucial for the proper functioning of β-oxidation. The peroxisomal CoA is again released, for instance, once the acetyl-CoA is fed into the glyoxylate cycle, resulting in CoA recycling. Like in humans and yeast, plant peroxisomes possess several acyl-CoA thioesterases to ensure optimal flux through β-oxidation. They have been proposed to regulate the availability of free CoA in the peroxisomal matrix, by releasing CoA from acyl-CoA esters [109]. In addition, a role in maintaining the peroxisomal CoA has also been reported for the putative Nudix hydrolases of plant peroxisomes [110]. Since the CoA biosynthesis and salvage pathway takes place outside peroxisomes [111], plant peroxisomes depend on the uptake of CoA, which might be mediated by a specific CoA transport protein. A mitochondrial carrier for the distribution of CoA has been identified and characterized in humans, yeast, and plants [43,75], but the knowledge about a peroxisomal CoA carrier is restricted to non-plant organisms. An ortholog of the peroxisomal CoA carrier SLC25A17 in humans and animals is still unknown in Arabidopsis. The peroxisomal NAD carrier PXN, however, showed transport activities for CoA in vitro [100]. However, due to its low affinity to CoA, it is unlikely that PXN is able to transport CoA under physiological conditions [37]. Biochemical and genetic analyses will demonstrate the existence of a peroxisomal CoA carrier in plants in the future.

## 5. Conclusions

The mitochondrial carrier family (MCF) is a large family of proteins present in all eukaryotic lineages, which are present in several other cellular compartments, including peroxisomes. The peroxisomal MCF-type proteins described so far in yeast, humans, and plants, cluster as a functional branch in phylogenetic analyses [18,110]. These carriers ensure a stable exchange of structurally related cofactor molecules, such as ATP, CoA, and NAD, which are required for the maintenance of peroxisomal reactions, in particular β-oxidation. A detailed characterization of the transport function is indispensable to understand their physiological role in peroxisomes. However, the exciting findings of the knockout mutant displayed that alternative routes with a so far unknown mechanism exist for the cofactor exchange across the peroxisomal membrane.

Beyond cofactors, the diverse anabolic and catabolic reactions in peroxisomes produce a large number of small hydrophilic metabolites, which have to be shuttled across the peroxisomal membrane. Peroxisomes appear to be permeable to small hydrophilic solutes. Experimental evidences suggest a nonselective channel responsible for this exchange, but the proposed protein could not yet be conclusively assigned to the observed channel activity [3,6,8]. This raises the question if other transport proteins might be involved in peroxisomal metabolite transfer. For instance, MCF represents a carrier family with a broad substrate spectrum, catalyzing the transport of small hydrophilic solutes, like amino acids, mono, di- and tricarboxylates, and an inorganic form of phosphate. It could be hypothesized that—beyond the cofactor uptake—MCF-type proteins might play a role in these transport processes. Until now, such an MCF protein has not yet been located in peroxisomes in any eukaryotic organisms.

## Figures and Tables

**Figure 1 biomolecules-10-01174-f001:**
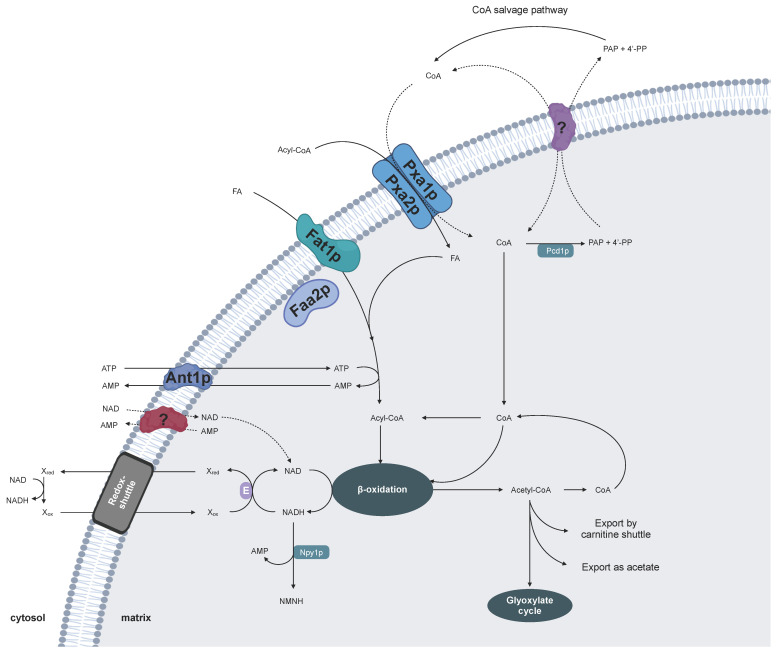
Model for the peroxisomal cofactor transport in yeast. Ant1p: peroxisomal ATP carrier 1; FA, fatty acid; Faa2p: peroxisomal acyl-CoA synthetase 2 (associated at the matrix side of the peroxisomal membrane); Fat1p: peroxisomal acyl-CoA synthetase, also known as peroxisomal fatty-acid transporter 1 (integral membrane protein); NMN(H): nicotinamide mononucleotide (reduced); Npy1p: peroxisomal NAD(H) diphosphatase 1, catalyzing the hydrolysis of NADH to AMP and NMN(H); PAP: adenosine 3′,5′-diphosphate; 4′-PP: 4′-phospho-pantetheine; Pcd1p: peroxisomal CoA diphosphatase 1, catalyzing the hydrolysis of CoA to PAP and 4′-PP; Pxa1p/Pxa2p: peroxisomal ATP-binding cassette (ABC) transporter, mediating the import of fatty acids as acyl-CoA esters; (?): proposed cofactor carrier proteins.

**Figure 2 biomolecules-10-01174-f002:**
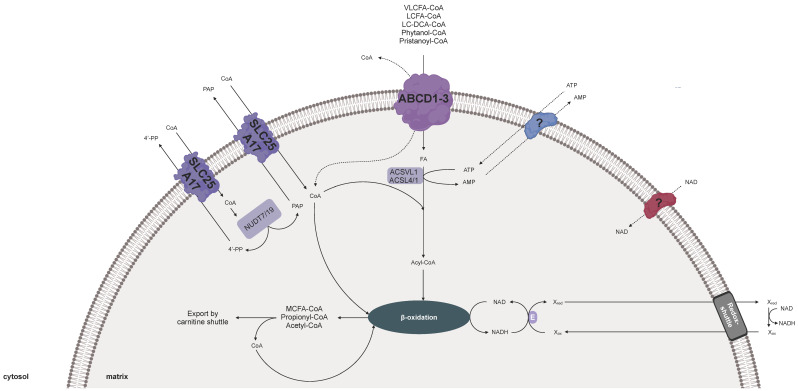
Model for the peroxisomal cofactor transport in human. ABCD1–3: ABC transporters of the subfamily D (ABCD) member 1–3, mediating the import of fatty acids and other β-oxidation-related substrates as CoA esters; ACSVL1: very-long-chain acyl-CoA synthetase 1 (associated at the matrix site of the peroxisomal membrane); ACSL1/4: long-chain acyl-CoA synthetase1 and 4 (associated at the cytosolic site of the peroxisomal membrane); FA, fatty acid; NUDT7/19: peroxisomal Nudix hydrolase 7 and 19, catalyzing the hydrolysis of CoA to PAP and 4′-PP.; LCFA-CoA: long-chain acyl-CoAs; LC-DCA-CoA: long-chain dicarboxyl-CoA; PAP: adenosine 3′,5′-diphosphate; 4′-PP: 4′-phospho-pantetheine; VLCFA-CoA: very long-chain acyl-CoAs; SLC25A17: solute carrier family 25 member 17, putative peroxisomal CoA carrier; (?): proposed cofactor carrier proteins.

**Figure 3 biomolecules-10-01174-f003:**
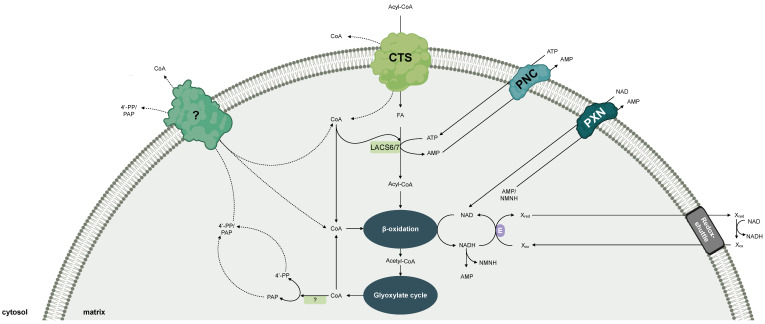
Model for the peroxisomal cofactor transport in plants. CTS: peroxisomal ABC transporter, mediating the import of fatty acids as acyl-CoA esters; FA: fatty acid; LACS6/7: long-chain acyl-CoA synthetase 6 and 7 (associated at the matrix site of the peroxisomal membrane); NMN(H): nicotinamide mononucleotide (reduced); PAP: adenosine 3′,5′-diphosphate; 4′-PP: 4′-phospho-pantetheine; PNC, peroxisomal ATP carrier; PXN, peroxisomal NAD carrier; (?): proposed cofactor carrier proteins.

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
