# Peer review of "Peroxisomal Cofactor Transport"

_biomolecules, 2020, doi:10.3390/biom10081174_

Round 1
Reviewer 1 Report
The review article by Plett et al. provides a very timely, readable, and comprehensive overview on cofactor import into peroxisomes, focussing on MCF proteins. I recommend this review for publishing in “Biomolecules” with minor changes. The following corrections and suggestions are meant to improve the article.
MAJOR POINTS
2 (Title) – “Mitochondrial carriers in peroxisomes” .
The only major points concern the title and the focus of this article. To the specialist it may be clear that “mitochondrial carriers” refers to “mitochondrial carrier family (MCF) proteins”. The general reader, however, is misled in the double sense: The article does not touch mitochondrial (carrier) proteins that are partially or temporarily localized to peroxisomes (as the title may suggest). Neither is the discussion confined to MCF exclusively. The inclusion of ABC transporters in all chapters may even suggest that these belong to the superfamily of MCF. So the authors should change the title to reflect the true focus of the article. I would find it more natural to speak of “cofactor transport”. The focus on MCF then becomes clear in the abstract.
These adjustments should affect the text not to lose focus. In addition, the authors may consider changing the title/focus to “metabolite transport”, as fatty acid transport is also discussed in the paper.
MINOR POINTS (line numbers)
55 “depends” change to “depend”
57 “have to be efficiently, still controlled exchanged” – consider rephrasing
70 clarify “non-covalently, organic cofactors”
71 explain ATP-requirement in beta-oxidation (to my understanding beta-oxidation does not require ATP)
84 explain “cofactor-couples import”
102 “MCF members catalyse a strict counter exchange” is overstretching the concept of catalysis, because there is no chemical reaction involved
120 “half-size ATP-binding cassette” – What does “half-size” refer to?
123 “prior” – change to “prior to”, or better “before”
130 “YPR128C” change to “Ypr128cp” (protein). There are several other instances, eg. on page 7 of the manuscript, where genes are stated in brackets. Imo this is not necessary, the protein names are sufficient. If the authors feel that systematic yeast names should be stated, they should do so for all yeast proteins mentioned.
141 (Figure1) – make sure, all abbreviations are explained in the figure legend (FFA)
156 “successful system” – reconsider expression
156 “subsequently” – change to “subsequent”
179 “Future research …” – sentence is not clear, please check
215 I do not agree that peroxisomes are absent in male germ cells. See eg: Dastig et al. Histochem Cell Biol 2011, vol 136, p 413.
192 “have been” – change to “has been”
204 NAD/ NADH “redox shuttles” – discussion should be linked to mammalian redox shuttles (see following comment)
223 (Chapter 3) and Figure 2 (line 280) – In the last couple of years several groups have identified lactate and malate dehydrogenases in the mammalian peroxisome. As they are also present in the cytosol, it is a plausible speculation that these enzymes support a metabolite shuttle for NAD. The authors should refer to these studies as they are in the focus of this review.
229 “likewise” – explain more clearly what is meant here
246 “for example” – not clear if example of ACBD or function of ACBD5
260 “does not dependent” – change to “does not depend” or “is not dependent”
313 change “ant” to “and”
323 spell “nutrient-rich”
324 change “in the” to “the”
366 spell “peroxisomal pool of free CoA” (if that is what is meant)
374 “?” – question mark not required as the question is not stated as such
397 “mechanism” – change to “mechanisms”
400 reference [76] - the discussion is not satisfactory here, because the reference is exclusively on a isocitrate dehydrogenase, see also comments line 223
432 change “in respect” to “with respect”
435 (Figure 3) “CoA” (second from above) should not point to “LACS6/7” but to “Acyl-CoA”; resolve the relationship of “CoA” (below) below with “PAP and “4’-PP”. Currently it looks like CoA is catalysing the interconversion of PAP and 4’-PP.
473 consider “are located” or “reside” instead of “occur”
499 insert “be” after “might”
546 “the loss-of-function mutant” – restate which mutant is meant here
553 “conclusive assigned” - change to “be conclusively assigned”
553 ”the proposed protein” – The authors are referring to PXMP2 here without mentioning it? A discussion of PXMP2 including references should be included in the Chapter 3.
564 „due to space limitations“ – Biomolecules, to my understanding, does not impose limitations to the number of cited items in a review article. So there is no need to exclude essential citations or to apologize to authors that are not cited.
575 (References) Issue numbers should not be stated in the references.
595 volume and page range missing
716 volume and page range missing
Author Response
Major points
We chose the title because the this special issue is about mitochondria, but we agree with the reviewer and changed the title to ‘Peroxisomal cofactor transport’, since the main focus of the manuscript is the transport of the cofactor molecules ATP, NAD and CoA. We did not change the title to ‘metabolite transport at the peroxisomal membrane’, because the manuscript excluded the peroxisomal channel proteins responsible for the exchange of other solutes.
Minor points
Introduction
Line 56 (55): We apologize and corrected the grammatical error.
Line 58 (57): We removed the words “have to be efficiently, still controlled”.
Line 73 (70): We removed the words “non-covalently, organic”.
Line 74-75 (71): The activation of fatty acids require ATP and CoA. This reaction is a perquisite for beta-oxidation and thus is part of this pathway. We rephrase the sentences and included ATP- and CoA-dependent fatty acid activation.
Line 88 (84): We rephrase it to ‘cofactor-coupled protein import’ to clarify this.
Line 146 (102): We exchanged the verb ‘catalyze’ to ‘mediate’.
Chapter 1
Line 164/213 (120): We deleted it. It is not important to explain it more detail how ABC transport proteins are structured, because the review is focusing on MCF proteins.
Line 167 (123): We changed ‘prior’ to ‘prior to’.
Line 174 (130): We agree and removed the yeast gene IDs in the manuscript.
Figure 1-3: We re-checked all abbreviations in the three figures and added FFA for ‘free fatty acid’ to the figure legend 1 and 3.
Line 219 156): We exchanged ‘successful’ into ‘suitable’.
Line 219 (156): We apologize and corrected the grammatical error.
Line 184-187 (179): The last two sentences were combined for a better understanding and reduce redundancy.
Line 192 (192): We could not find the entry ‘have been’ for changing it to ‘has been’. But we correct the grammatical error in line 319.
Line 239-241 (204/223): We mentioned additional NAD(P)-linked redox shuttles in the yeast chapter. The main focus here is the transporter that import NAD into peroxisomes. Once NAD is present in the peroxisomal matrix, the redox shuttles can be functional by transferring reducing equivalents across the membrane. These NAD(P)-linked shuttles do not directly mediate the transport of NAD or NADH. To clarify this, we included these redox shuttles to figure 1.
Chapter 3
Line 312 (215): We apologize for this oversight and removed ‘male germ cells’.
Line 316 (229): We deleted the word ‘likewise’
Line 332-335 (246): We rephrase the sentences and removed ‘for example’.
Line 392 (260): We apologize and corrected ‘does not dependent’ to ‘is not dependent’.
Line 467 (313): We apologize and corrected the grammatical error.
Line 477 (323): We corrected ‘nutrient rich’ to ‘nutrient-rich’.
Line 478 (324): We apologize and corrected the grammatical error.
Line 540 (366): We changed ‘peroxisomal pool of free CoA’ to “peroxisomal CoA pool’ for a better understanding.
Line 573 (374): We deleted the question mark.
Line 595 (397): We apologize and corrected the grammatical error.
Line 595-598 (400): We cited the review ‘Metabolic interplay between peroxisomes and other subcellular organelles including mitochondria and the endoplasmic reticulum.’ by Wanders et al [Ref No. 4], discussing the existent redox shuttles of mammalian peroxisomes.
Chapter 4
Line 672 (432): We apologize and corrected the grammatical error.
Figure 3 (435): We agree and changed it in the figure 2.
Line 738 (473): We changed ‘occur’ to ‘are located.
Line 739 (499): We apologize and corrected the grammatical error.
Line 787 (546): We changed ‘loss-of-function’ to ‘deletion of this gene’.
Conclusion
Line 851: We changed ‘loss-of-function’ to ‘knockout’.
Line 858 (553): We changed ‘conclusive assigned’ to ‘be conclusively assigned’.
Line 859 (553): We referring to a review describing the putative role of the peroxisomal channel protein in eukaryotes. Our manuscript addresses the MCF proteins and thus we excluded the Sym1/Mvp17/Pxmp2/PMP22 protein family and its speculative function in peroxisomal metabolite transport.
Acknowledgment
Line 881 (564): We apologize and deleted the phrase ‘due to space limitations’.
References
We apologize and deleted the issue number for every cited reference.
Line 916 (595): We apologize and added the volume and page number to this reference.
Line 1177 (716): We apologize and added the volume and page number to this reference.
Reviewer 2 Report
The manuscript by Plett et al. entitled: Mitochondrial carriers in peroxisomes is a carefully conducted review adequately describing the current state of knowledge about mitochondrial carriers in peroxisomes with emphasis on the yeast Saccharomyces cerevisiae, humans and plants. The review is a well-balanced mix between data from the group of the senior author herself as well as work by others, partially overlaps with a recent review from the same set of authors but provides sufficient new information to warrant publication.. I only have limited points of criticism which include:
(1.) Lines 68-69: I suggest to include the original references here rather than referring to 3 reviews;
(2.) Lines 82-88: it may be wise to emphasize that the protein-coupled import of cofactors may only work out for cofactors tightly bound to enzyme proteins as is the case for acyl-CoA oxidase as well HACL but this does apply to cofactors such as NAD(H), CoASH, and ATP. Please make this clear in the text.
(3.) Line 118: Ref.26 should be 27.
(4.) Line 126: This line reads: " This intraperoxisomal esterification is catalyzed by peroxisomal acyl-CoA synthetase Faa2p and/or bifunctional fatty acid transporter Fat1p at the peroxisomal membrane". I agree that Faa2p is definitely localized in the peroxisomal lumen and does react with intraperoxisomal ATp and CoASH but I am unsure about Fat1p. Please specify whether it has been established or not that Fat1p truly reacts with intraperoxisomal ATP, and CoASH. Give references here!
(5.)Figure 1: I have difficulty with this Figure for the following reasons: in the upper right part of the Figure an unidentified carrier is drawn which is supposed to import CoASH in exchange for 4'-PP. However, due to the import of CoASH by the pxa1/Pxa2-heterodimer, there is already import of CoASH which explains why I would think that there should actually be an export system for CoASH rather than an import system! Please explain. One more thing: acetyl-CoA can indeed escape from the peroxisome as acetylcarnitine but also as acetate. Please correct.
(6.) Line 242: wrong reference;please correct.
(7.) Lines 359-372: this section deals with the problem of the activation of branched-chain fatty acids. It is important to emphasize here that pristanic acid is derived from two sources one being direct from dietary sources and the other indirect as pristanic acid is generated from phytanic acid by alpha-oxidation. This is relevant because in the latter case pristanic acid is generated INSIDE peroxisomes and is most likely activated intraperoxisomally by ACSVL1 whereas the pristanic acid coming from the diet is most likely activated outside peroxisomes and imported via ABCD3. Please read the relevant literature to this point (see Jansen et al. (2001) BBRC 283,674-679 and Wanders et al.(2011) BBA 1811,498-507) and rewrite the text.
(8.) Lines 392-394:it has been clearly established that at least in human peroxisomes none of the enzymes of the mevalonate pathway claimed to be peroxisomal by Krisans and coworkers, are in fact localized in peroxisomes. In fact they are all cytosolic! Please inspect your Ref.41 to this point and correct the text.
Author Response
Minor points
(1.) Lines 72 (68-69): We included the original references for the known peroxisomal cofactor carrier.
(2.) Lines 82-83 (83-84): We clarified in the manuscript that the protein-coupled import of cofactor may play a role for cofactors that are tightly bound to the enzymes, such FAD and TPP.
(3.) Line 127 (118): We apologize and corrected the reference mistake.
(4.) Line 170 (126): The reference Nr. 35 cited here demonstrated using genetic yeast mutants that Fat1p is responsible for the peroxisomal activation of long-chain fatty acids (C22:0) like Faa2p and its function is associated with the ABC transporter Pxa1p-Pxa2p. Because of this experimental evidence we did not change this sentence.
(5.) Figure 1: We agree and corrected the figure 1.
(6.) Line 242 (269): We apologize and corrected the reference mistake.
(7.) Lines 514-524 (359-372): We apologize the confusion and rephrased the text.
(8.) Line 591 (392-394): We apologize and removed cholesterol biosynthesis as ATP-dependent pathway for human peroxisomes.